# EMERGENCE OF EXPLORATION IN POLICY GRADIENT REINFORCEMENT LEARNING VIA RESETTING

## ABSTRACT

In reinforcement learning (RL), many exploration methods explicitly promote stochastic policies, e.g., by adding an entropy bonus. We argue that exploration only matters in RL because the agent repeatedly encounters the same or similar states, so that it is beneficial to gradually improve the performance over the encounters; otherwise, the greedy policy would be optimal. Based on this intuition, we propose ReMax, an objective for RL whereby stochastic exploration arises as an emergent property, without adding any explicit exploration bonus. In ReMax, an episode is modified so that the agent can reset to previous states in the trajectory, and the agent's goal is to maximize the best return in the trajectory tree. We show that this ReMax objective can be directly optimized with an unbiased policy gradient method. Experiments confirm that ReMax leads to the emergence of a stochastic exploration policy, and improves the performance compared to RL with no exploration bonus.

## 1 INTRODUCTION

Exploration is widely studied in reinforcement learning (RL) (Sutton & Barto, 2018) (see App. A for an extended overview). The most popular method of exploration is to explicitly promote a stochastic policy by maximizing the entropy in addition to the rewards (Williams, 1992; Ziebart et al., 2008; Mnih et al., 2016; Haarnoja et al., 2018). We note that it is non-obvious why one should add such an entropy bonus—the objective of RL is only to maximize the rewards. Such exploration methods are only retrospectively justified as they improve the performance of the algorithms. In our article, we propose a method that, paradoxically, promotes exploration by greedily maximizing the rewards.

The motivation of our method is the following: we suppose that exploration is vital in RL because the agent, intentionally or unintentionally, visits the same (or similar) state repeatedly; exploration allows the gain of some valuable information for making a better decision on the next visit to the same state. However, it has no value if the agent would never encounter the same state.

Based on this observation, we propose a new objective function for RL called ReMax that encourages exploration in a novel way. Briefly, the ReMax objective is computed as follows: while interacting with the environment, in addition to taking usual actions, the agent may choose to reset to a previously visited state in the trajectory up to some limited number of times; then, after the interaction, the value of the ReMax objective is computed as the sum of the rewards along the best trajectory.

The crucial difference between our approach and previous ones is that, while most previous approaches explicitly set the goal of obtaining a stochastic exploratory policy via an exploration bonus (e.g., state-visitation bonus or entropy bonus), in our approach, such an exploratory policy is not the explicit goal, but the optimization of the ReMax objective naturally results in an exploratory policy.

We note that several previous studies successfully utilized resetting. For example, Go-Explore (Ecoffet et al., 2021), which achieved impressive results on the well-known hard-exploration problem Montezuma's Revenge, utilized resetting to rarely visited states, but also AlphaGo (Silver et al., 2016) used Monte-Carlo tree search that can be regarded as a kind of resetting. In practical RL problems, resetting is often possible, such as when we have access to the environment simulator (like Go). Also, even if such simulator access is not available, we can use powerful model-based RL (MBRL) methods, e.g., DreamerV2 (Hafner et al., 2021), and use resetting in simulations with the learned model.

The main objective of this article is to test our hypothesis that ReMax leads to the emergence of a stochastic exploratory policy. To this end, we perform three phases of experiments:

- **Step 1.** We illustrate the main idea and demonstrate that optimizing the ReMax objective causes a stochastic policy in a simple bandit task (Sec. 3). This experiment was non-conclusive as the emergence of the stochastic policy relied on the partial observability of the environment.
- **Step 2.** To overcome the limitation of the previous step, we demonstrate that, by optimizing the ReMax objective, a stochastic policy emerges even in a deterministic maze environment, where optimizing the regular RL objective causes the policy to become deterministic and the learning to stop (Sec. 5). The limitation here is that the example relied on a simple model parameterization.
- **Step 3.** To make the scenario of the maze experiment more realistic, we modify the maze to represent the observations by images, and use a neural network function approximator (Sec. 6). This experiment indicates that the failure of the regular RL and the emergence of exploration happen in a practical deep RL scenario, even in a deterministic environment.

Finally, we showed that ReMax can promote exploration in modern policy gradient algorithms, such as A2C (Mnih et al., 2016), and improve the performance in MinAtar (Young & Tian, 2019), a simplified version of the Arcade Learning Environment (Bellemare et al., 2013)(Sec. 8.1). We believe ReMax is a viable competitor to classical stochastic exploration approaches, such as entropy bonuses.

## 2 PRELIMINARIES

**Notation.** We consider an episodic Markov decision process (MDP) $\mathcal{M}$, defined as a tuple $(\mathcal{S}, \mathcal{A}, P, r, \rho_0, T)$, where the state space $\mathcal{S}$, the action space $\mathcal{A}$ are discrete and $T$ is a finite horizon. The initial state $s_0 \in \mathcal{S}$ follows the distribution $\rho_0 : \mathcal{S} \to [0, 1]$, and the state transition kernel $P : \mathcal{S} \times \mathcal{A} \times \mathcal{S} \to [0, 1]$ defines the state transition probability from the current state $s \in \mathcal{S}$ to the next state $s' \in \mathcal{S}$ after the action $a \in \mathcal{A}$ is taken. The reward function $r : \mathcal{S} \times \mathcal{A} \to [r_{\min}, r_{\max}]$ determines the immediate reward given the state, $s$, and action, $a$. At each state, $s$, the agent can take a legal action $a \in \mathcal{A}(s) \subset \mathcal{A}$, where $\mathcal{A}(s)$ are the legal actions at state $s$. The agent acts following a parameterized policy $\pi_\theta : \mathcal{S} \times \mathcal{A} \to [0, 1]$ with the goal of maximizing the rewards. The trajectory $\tau := (s_0, a_0, \ldots, s_T)$ is the sequence of state-action pairs from the current episode: $\tau \sim \rho_\pi(\tau)$ where $\rho_\pi(\tau) := \rho_0(s_0) \prod_{t=0}^{T-1} \pi(a_t|s_t) P(s_{t+1}|s_t, a_t)$. Note that $s_T$ is the terminal state. The RL objective is to maximize the expected return $J_{\mathrm{RL}}(\pi) := \mathbb{E}_{\tau \sim \rho_\pi}[\mathcal{R}(\tau)]$, where $\mathcal{R}(\tau) = \sum_{t=0}^{T-1} r(s_t, a_t)$.

**Policy gradient methods.** In this study, we focus on the policy gradient (PG) method, which directly optimizes a parameterized policy $\pi_\theta$ via gradient ascent. The policy gradient theorem (Sutton et al., 1999) provides an expression of the PG, $\nabla_\theta J_{\mathrm{RL}}(\pi_\theta)$, amenable for estimation. In particular, we use REINFORCE (Williams, 1992) as the simplest PG method, whose gradient estimator is given by $\hat{g} := \sum_{t=0}^{T-1} \nabla_\theta \log \pi_\theta(a_t|s_t)(\mathcal{R}(\tau) - b_t)$, where $b_t$ is a constant baseline for variance reduction. This estimator is unbiased: $\nabla_\theta J_{\mathrm{RL}}(\pi_\theta) = \mathbb{E}_\tau[\hat{g}]$. One may also average a batch of $N$ gradient estimates from different trajectories, $\sum_{i=1}^{N} \frac{1}{N} \hat{g}_i$. A common baseline is $b_t = \sum_{i=1}^{N} \frac{1}{N} \mathcal{R}(\tau_i)$, the average of the returns in the batch. Another common method to reduce the variance is using the future return $\mathcal{R}_t(\tau) := \sum_{h=t}^{T-1} r(s_h, a_h)$, that only includes the rewards following the action; this maintains the unbiasedness of the estimator. An important property of PG methods—and part of the reason we focus on them—is that they remain unbiased even when the system is a POMDP (partially observable MDP), i.e., unobservable hidden states characterize the state transitions.

## 3 STEP 1: BANDIT PROBLEM EXAMPLE

In the first step of our 3-stage experiment, we illustrate the core idea behind our ReMax objective. Through a simple randomized bandit task, we explain the principle of why a stochastic policy is optimal under the ReMax objective; thus, leading to the emergence of exploration.

**Problem.** There are two arms, indexed by $0$ and $1$. At the beginning of each episode, one arm is chosen as an unobservable "correct" arm $z \in \{0, 1\}$. The correct arm $z \in \{0, 1\}$ is randomly chosen according to a Bernoulli distribution with probability $q = 0.75$. In each episode, the agent plays only one arm $a \in \{0, 1\}$. Playing the correct arm (i.e., $a = z$) gives the return $1$ and $0$ otherwise: $\mathcal{R}(z, a) = \mathbb{I}_{z=a}$, where $\mathbb{I}_e$ takes $1$ if $e$ is true and $0$ otherwise. Under the usual RL objective, which maximizes expected return $\mathbb{E}_{z,a}[\mathcal{R}(z, a)]$, the optimal policy is deterministic, taking action $a = 1$ with probability $1$, which yields a maximum expected return $0.75$.

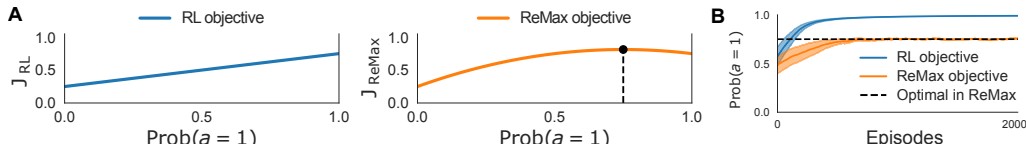

**Figure 1: Bandit problem example. (A)** A comparison of two objective functions: RL objective *(Left)* and ReMax objective with $K = 2$ *(Right)*. The black dotted line indicates the optimal policy. **(B)** Empirical results of optimizing the policies with the RL objective and the ReMax objective.

**ReMax objective.** We define our ReMax objective on this bandit problem as:

$$J_{\text{ReMax}}^{(K)}(\pi) := \mathbb{E}_z\left[\mathbb{E}_{a^{(1)},\ldots,a^{(K)}}\left[\max_{k\in\{1\ldots,K\}}\mathcal{R}(z,a^{(k)})\Big|z\right]\right]. \tag{1}$$

In this objective *the agent has $K$ chances to choose an arm, and the best of those $K$ returns is defined as the value to optimize.* For the regular RL objective, we saw that a deterministic policy was optimal; however, for the ReMax objective, a stochastic policy is optimal instead. To understand this intuitively, consider that pulling the same arm multiple times does not affect the ReMax objective, while pulling both arms guarantees pulling the correct arm, giving the return $\max\{\mathcal{R}(z,0),\mathcal{R}(z,1)\} = 1$. We can analytically compute the expected return in this augmented problem (App. B). The solutions are in Fig. 1 (A) where we compare the ReMax objective ($K = 2$) and the RL objective. We can confirm that $\pi(a = 1) = 0.75$ maximizes the ReMax objective (shown as the black dotted line), and the optimal policy under the ReMax objective is exploratory.

**Experiments.** We also experimentally confirmed that, when trained to maximize the ReMax objective, a direct policy search algorithm converges to the stochastic policy rather than a deterministic one. We trained a policy with only one parameter $\theta \in \mathbb{R}$. The probability of selecting action 1 is defined as $\pi_\theta(a = 1) = \sigma(\theta)$, where $\sigma$ is the sigmoid function. We initialized $\theta$ so that $\pi_\theta$ distributes uniformly over $[0, 1]$. We used $K = 2$ in this experiment. Given a sample $(z, a^{(1)}, a^{(2)})$, we updated $\theta$ using $\Delta\theta = -\alpha\left(\sum_{k=1}^2 \nabla_\theta \log \pi_\theta(a^{(k)})\right)\left(\max_{k'\in\{1,2\}}\{\mathcal{R}(z, a^{(k')})\}\right)$, where $\alpha = 0.01$ is the step-size parameter. Fig. 1 (B) shows the results of this training procedure. Each line shows an average performance of 10 runs, and the shaded area indicates the standard error. We can see that the policy converges to the deterministic one under the standard RL objective, whereas the policy converges to the optimal stochastic policy under the ReMax objective as expected.

**Discussion.** The key insight from this bandit problem example is that *if the agent can repeat the decision-making, seeking the best result in the same state, the optimal policy may be stochastic.* This result relied on the uncertainty due to the hidden state $z$. It may be unclear whether the same result will hold up in typical fully observable MDPs. However, interestingly, we will show that a stochastic policy emerges even in a fully observable deterministic MDP (Sec. 5), where we argued that there is an implicit uncertainty due to function approximation and distribution shift during learning.

## 4 ReMax with resetting in MDPs

In preparation of stages 2 and 3 of our experiments (Secs. 5, 6), we lay the foundations of using ReMax in MDPs. In the bandit example (Sec. 3), the agent had $K$ chances to play at the given state. However, in RL, it is not practical to act $K$ times at each state, especially when the environment is large. To optimize the ReMax objective in a practical way, we utilize *resetting* and define a *resettable MDP* (ReMDP), where the agent has a special action to "jump" back to previously visited states in the episode, and the *trajectory tree*, a tree constructed by the state-action pairs in a ReMDP episode. Fig. 2 shows the idea of resetting and the trajectory tree. We define our ReMax objective over the ReMDP (Sec. 4.1) and also describe a PG method that optimizes the objective (Sec. 4.2).

### 4.1 ReMax objective in MDPs

**Resettable MDP.** We define a ReMDP $\mathcal{M}_{\text{Re}} = (\mathcal{S}, \mathcal{A}_{\text{Re}}, P_{\text{Re}}, r, \rho_0, T)$ by extending the action space and state transition kernel of the original MDP $\mathcal{M} = (\mathcal{S}, \mathcal{A}, P, r, \rho_0, T)$ as follows: the action $u \in \mathcal{A}_{\text{Re}}$ is an element of $\mathcal{A}_{\text{Re}} := \mathcal{A} \bigcup \mathcal{X}$, where $x \in \mathcal{X}$ indicates a reset action to the *target state*

$s_{\mathrm{Re}}(x) \in \mathcal{S}$. If a reset action is chosen ($u \in \mathcal{X}$), the state immediately transitions to $s_{\mathrm{Re}}(u)$:

$$\mathrm{P}_{\mathrm{Re}}(s'|s,u) := \begin{cases} \mathbb{I}_{s'=s_{\mathrm{Re}}(u)} & \text{if } u \in \mathcal{X} \\ \mathrm{P}(s'|s,a) & \text{if } u = a \in \mathcal{A} \end{cases}.$$

However, as our motivation was to repeat decisions in the same state, *the target states of resetting are limited to previously visited states in the episode*: the legal actions at $s_t$ are $\mathcal{A}_{\mathrm{Re}}(s_t) := \mathcal{A}(s_t) \bigcup \mathcal{X}(s_t)$ such that for any $x \in \mathcal{X}(s_t)$, $s_{\mathrm{Re}}(x) \in \{s_0, \ldots, s_{t-1}\}$ holds.

**Trajectory tree.** Now, we can sample $\mathcal{T} := (s_0, u_0, \ldots, s_T)$, a trajectory on the ReMDP, $\mathcal{M}_{\mathrm{Re}}$. We call $\mathcal{T}$ a *trajectory tree* because we can construct a tree, whose nodes correspond to the states $s \in \mathcal{S}$, and edges correspond to the actions $u = a \in \mathcal{A}$. See Fig. 2 for an example trajectory tree, $(s_0, u_0, \ldots, s_7)$. The initial state $s_0$ is the root node, while the states where a reset happened and the terminal states are the leaf nodes.

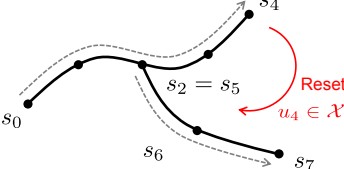

**Figure 2: Trajectory tree example.**

**ReMax return.** Given a trajectory tree $\mathcal{T}$, we define our ReMax return $\mathcal{R}_{\mathrm{ReMax}}(\mathcal{T})$. Here, we assume that the trajectory tree $\mathcal{T}$ has $K$ leaf nodes. For $k \in \{1, \ldots, K\}$, we define the trajectory path $\tau^{(k)}$ as the subsequence of $\mathcal{T}$ consisting of the states and actions in the path from the root node to the $k$-th leaf. For example, the trajectory tree in Fig. 2 has two trajectory paths $\tau^{(1)} = (s_0, u_0, \ldots, s_4)$ and $\tau^{(2)} = (s_0, u_0, \ldots, s_2 = s_5, u_5, \ldots, s_7)$. We define the ReMax return on the trajectory tree by

$$\mathcal{R}_{\mathrm{ReMax}}(\mathcal{T}) := \max_{k \in \{1,\ldots,K\}} \mathcal{R}(\tau^{(k)}), \tag{2}$$

where, $\mathcal{R}(\tau^{(k)})$ is the return along the path $\tau^{(k)}$ defined as in conventional RL.

**ReMax objective.** Our proposed objective is to maximize the expected ReMax return:

$$J_{\mathrm{ReMax}}(\pi) := \mathbb{E}_{\mathcal{T}}\big[\mathcal{R}_{\mathrm{ReMax}}(\mathcal{T})\big]. \tag{3}$$

Note that if no resetting occurs, $J_{\mathrm{ReMax}}$ reduces to $J_{\mathrm{RL}}$. It is worth mentioning that if the MDP is deterministic, there exists a deterministic policy $\pi^* : \mathcal{S} \to \mathcal{A}$ that maximizes both $J_{\mathrm{RL}}$ and $J_{\mathrm{ReMax}}$. This is obvious as an optimal policy for $J_{\mathrm{RL}}$ also maximizes $J_{\mathrm{ReMax}}$ because resetting cannot increase the ReMax return if all of the chosen actions were optimal. Interestingly, *while there exists a common deterministic optimal policy, optimizing the ReMax objective enhances exploration during training in a deterministic environment*. We describe this phenomenon in Sec. 5.

## 4.2 ReMax policy gradient method

We propose to optimize the ReMax objective using policy gradients. As the simplest realization of the ReMax PG method, we consider REINFORCE. Given a trajectory tree $\mathcal{T}$, our gradient estimator is

$$\hat{g}_{\mathrm{ReMax}} := \sum_{t=0}^{T-1} \nabla_\phi \log \pi_\phi(u_t|s_t)\big(\mathcal{R}_{\mathrm{ReMax}}(\mathcal{T}) - b_t\big), \tag{4}$$

where $\pi_\phi : \mathcal{S} \times \mathcal{A}_{\mathrm{Re}} \to [0,1]$ is the parameterized policy to optimize, and $b_t$ is a baseline for variance reduction (Sec. 2). This estimator is unbiased: $\mathbb{E}_{\mathcal{T}}[\hat{g}_{\mathrm{ReMax}}] = \nabla_\phi J_{\mathrm{ReMax}}(\pi_\phi)$.

**Policies in this study.** In ReMDPs, agents must select actions from the extended action space $\mathcal{A}_{\mathrm{Re}} = \mathcal{A} \bigcup \mathcal{X}$. In this study, we decompose the policy $\pi_{\mathrm{Re}} : \mathcal{S} \times \mathcal{A}_{\mathrm{Re}} \to [0,1]$ into three independent components: the policy of the original MDP $\pi_\mathcal{A} : \mathcal{S} \times \mathcal{A} \to [0,1]$, the policy of *where to reset to* $\pi_\mathcal{X}(u|s) : \mathcal{S} \times \mathcal{X} \to [0,1]$, and the policy of *whether to reset* $y \sim \eta(y|s)$, where $y \in \{0,1\}$ is a Boolean variable: 1 means to reset, and 0 means to act in the MDP. Using these components, we have

$$\pi_{\mathrm{Re}}(u|s,y) = (1-y)\,\pi_\mathcal{A}(a|s) + y\,\pi_\mathcal{X}(x|s). \tag{5}$$

As our main focus in this study is the emergence of exploration from our ReMax objective, we only employ simple rule-based policies for $\pi_\mathcal{X}$ and $\eta$. We show that, even with such deterministic reset policies, the policy for the original MDP $\pi_\mathcal{A}$ becomes stochastic. Each realization of $\pi_\mathcal{X}$ and $\eta$ is described in the corresponding experimental setup section. We parameterize $\pi_\mathcal{A}$ as $\pi_\theta$. As the reset policy $\pi_\mathcal{X}$ is non-parameterized, we can rewrite Eq. 4 as

$$\hat{g}_{\mathrm{ReMax}} = \sum_{t|u_t=a_t \in \mathcal{A}} \nabla_\theta \log \pi_\theta(a_t|s_t)\big(\mathcal{R}_{\mathrm{ReMax}}(\mathcal{T}) - b_t\big), \tag{6}$$

where $t|u_t = a_t \in \mathcal{A}$ is an abbreviation of $t \in \{t|u_t = a_t \in \mathcal{A}\}$ and $\pi_\theta$ is the parameterized policy for the original MDP. One may use more sophisticated reset policies with search algorithms or trainable models for better performance. We leave such improved reset policies for future work.

**Reset Policy Gradient Theorem.** Finally, recall that in standard PG methods, one only needs to consider the rewards following an action in the PG, while ignoring the rewards obtained before the current time-step, as the action has no effect on what happened in the past. This is not the case in the reset PG method—the state may be reset to the past, cancelling out previously received rewards. Thus, we may have to consider the full return over the trajectory tree at each time-step. One wonders whether a similar PG theorem could be derived for the case with resets; whether some of the rewards in the return could be deleted while still guaranteeing unbiasedness. A sufficient condition for unbiasedness is formalized in the theorem below. Note that we provide this theorem for completeness and for a few justifications, but do not incorporate it in our algorithms.

**Theorem 1** (Reset Policy Gradient Theorem). *Denote $\tau^*$ is the optimal trajectory in the tree, $\mathcal{T}$, so that $\mathcal{R}_{\text{ReMax}}(\mathcal{T}) = \mathcal{R}(\tau^*) = \sum_{(s,a) \in \tau^*} r(s, a)$. Moreover, denote $\tau_{\text{fixed}}(s_t)$ is a subsequence $\tau_{\text{fixed}}(s_t) \in \tau^*$, starting at $s_0$ and ending at $s_h$, $h < t$, s.t. for all possible trajectories with $k \geq t$, and all $s \in \tau_{\text{fixed}}(s_t)$ we have $s \notin \mathcal{X}(s_k)$, where $\mathcal{X}(s_k)$ is the set of admissible states to reset to in state $s_k$. In other words, $\tau_{\text{fixed}}(s_t)$ is the set of states in the optimal trajectory to which it is impossible to reset to when starting at state $s_t$, at any point following time-step, $t$. Then we have*

$$\nabla_\theta \mathbb{E}\left[\mathcal{R}_{\text{ReMax}}(\mathcal{T})\right] = \mathbb{E}\left[\sum_{t|u_t = a_t \in \mathcal{A}} \nabla_\theta \log \pi_\theta(a_t|s_t)\left(\mathcal{R}_{\text{ReMax}}(\mathcal{T}) - \mathcal{R}(\tau_{\text{fixed}}(s_t)) - b_t\right)\right] \quad (7)$$

*Proof.* See App. D. $\qquad\square$

## 5 STEP 2: ReMax ON A DETERMINISTIC TABULAR MDP

In the bandit task (Sec. 3), the emergence of exploration relied on the uncertainty of the environment. To resolve this limitation, in step two of our experiments, we show that exploration also emerges in a deterministic MDP. We consider a tabular maze environment that we call the *biased maze* (Fig. 3). Though this maze has no *explicit* uncertainty or hidden state like the bandit task, we argue that function approximation and generalization leads to an *implicit* uncertainty, and we show that optimizing the ReMax objective promotes a stochastic exploratory policy even in deterministic MDPs.

**Problem.** The maze is a deterministic MDP with admissible actions 0 or 1 in each state. The agent receives a reward of 1 for each step forward. There is only one correct path in the maze (the red line in Fig. 3), and as long as the agent chooses the correct action, it can continue to move forward up to a maximum of 1000 steps. Once the agent chooses the wrong action, the maze will terminate after one or two steps, depending on whether the following action is also wrong (or not). We designed the maze such that the correct action is 1 in 75% of the states. In this sense, the maze is *biased* as action 1 tends to lead a higher return.

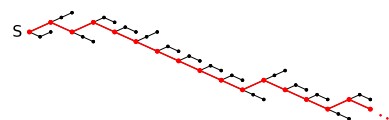

**Figure 3: Biased maze example.**

**Setup.** We train a policy $\pi(a = 1|s_i) = \sigma(w_i + c)$, where $i$ indicates the state index, each state $i$ has a corresponding local parameter $w_i$, and $c$ is a global scalar bias term shared by all states. We expect $w_i$ to extract the local information at the $i$-th state and $c$ to learn the global information over the states. We compared training the policy with the standard REINFORCE and with the ReMax version. In this experiment, the resets happen when the agent reaches a terminal state ($\eta(y = 1|s) = 1$ if $s$ is terminal and 0 otherwise), and the agent always resets back by $m = 2$ steps. We used the SGD optimizer. We estimate the gradient from a batch of 16 trajectory tree samples, and the baseline $b_t$ is calculated from the batch mean. For a fair comparison of the two algorithms, separate from the training episodes, we include *evaluation episodes* that are not used for training the policy. In the evaluation episodes, we use greedy action selection, as commonly done in previous work (Haarnoja et al., 2018; Hafner et al., 2021). We emphasize that no resetting is used in the evaluation episodes.

**Hypothesis.** *Biased maze* is a fully observable deterministic MDP. However, we claim that, during training, there exists uncertainty when the agent reaches a previously unseen new state. At each new state, the agent does not know whether action 1 or 0 is correct. Moreover, based on the structure of the

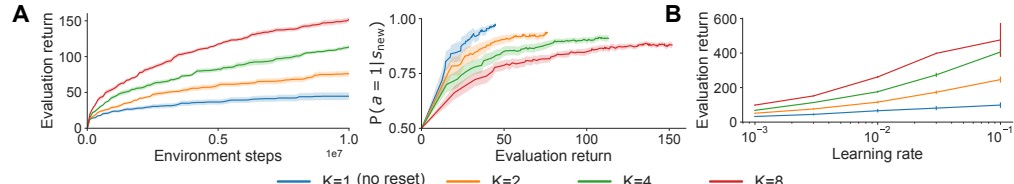

**Figure 4: Biased maze results. (A)** Evaluation return and average probability of selecting action 1 at a new (unseen) state with different $K$. The evaluation return is computed without resetting, even if $K > 1$. The shaded area indicates the standard error. **(B)** Evaluation return for different learning rates at 10M steps. The bar indicates the standard error.

maze, action 1 is preferable as it has a higher probability of being correct. As our agent can control the "prior" policy at unseen states using the bias parameter $c$, we hypothesize that ReMax should promote exploration even in such a deterministic MDP.[1] We note that such "implicit" uncertainty in fully observable MDPs has been previously studied in relation to generalization (Ghosh et al., 2021).

**Results.**    Fig. 4 (A) shows the results with the learning rate 0.003. The standard REINFORCE algorithm ($K = 1$), quickly converged to a suboptimal policy that deterministically chooses action 1 at an unseen state, and the evaluation return stopped increasing. On the other hand, ReMax ($K > 1$) promoted more exploratory policies. We also see that larger $K$ lead to more exploration and higher evaluation returns. These trends were consistent with other reasonable learning rates. Moreover, the regular REINFORCE cannot be greatly improved even when tuning the learning rate (Fig. 4 B). Finally, we also performed ablation studies in Sec. 7 and App. C.2 showing that not only the resetting, but also the maximization is important for the emergence of stochasticity, and improved performance.

**Discussion.**    Here, we showed that ReMax promotes exploration even in a deterministic MDP; however, this phenomenon relied on a shared parameter $c$. One may wonder whether this setup is realistic. In step three of our experiments (Sec. 6), we show that $c$ can be replaced with typical parameterized function approximator policies in RL, and demonstrate the consistency of the phenomenon.

## 6   STEP 3: REMAX ON A DETERMINISTIC MDP WITH VISUAL INPUTS

In the previous *biased maze* experiment, the model was too simple and it remained unclear whether exploration would emerge in a more practical scenario. In the third step of our experiments, we demonstrate that optimizing the ReMax objective promotes exploration even in a practical scenario with neural network function approximators, which have been proven successful in visual input environments (Mnih et al., 2015; Silver et al., 2016). For this purpose, we introduce the *MNIST maze* environment.

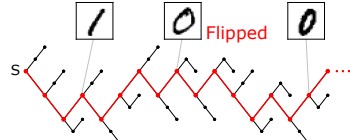

**Figure 5: MNIST maze example.**

**Problem.**    Inspired by Elfwing et al. (2016) we consider an *MNIST maze*, a modified version of the biased maze (Fig. 5). This maze is also a deterministic MDP but has visual inputs. In the maze, the agent can observe an MNIST image (LeCun et al., 1998) as a hint in addition to the state index. Each MNIST image is zero or one, indicating the correct action at the state. Unlike the *biased maze* there are an equal number of ones and zeros; however, the MNIST image hints are wrong with probability 0.25. Note that this flip does not change between the different episodes. The agent may solve the maze efficiently by generalizing the information in the images. However, if the agent blindly trusts the hints, it will fall into a suboptimal policy that deterministically follows the hints.

**Setup.**    We use the same setup as the biased maze experiment unless stated otherwise. Our neural network is a multilayer perceptron (MLP) with one hidden layer with 128 units, followed by a sigmoid activation function. We denote this MLP as $f_\phi$, where $\phi$ are the parameters of the MLP. The MLP takes the image, $\text{img}_i$, at state $i$ as an input and produces the output $y_i = f_\phi(\text{img}_i) \in \mathbb{R}$. The policy is $\pi(a = 1 | s_i) = \sigma(w_i + d y_i)$, where the $d$ hyperparameter controls the contribution from

---

[1]Note that without the bias parameter $c$, the agent cannot control the "prior" at unseen states. See App. C for the results without the bias term.

the MLP. We trained both the state-wise parameter $w$ and MLP parameters $\phi$ simultaneously. We tuned the learning rate and $d$ hyperparameters in a different environment setting, where the hint flip probability was zero. The chosen learning rate and $d$ are $0.003$ and $0.01$, respectively. See App. E for the validation experiments regarding the hyperparameters.

**Results and discussion.** Fig. 6 shows the results. Like in the biased maze results, when $K = 1$, the policy quickly became deterministic, trusting the MNIST image hints too much in new states, and the learning stopped. On the other hand, ReMax ($K > 1$) promoted exploration and showed better performance. Thus, we have demonstrated that exploration emerges even in practical scenarios that utilize neural network function approximators. Finally, we will add implementation tricks, and demonstrate the feasibility of using ReMax to promote exploration in modern PG algorithms (Sec. 8).

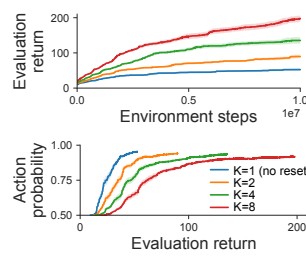

**Figure 6:** *MNIST maze* **results.**

## 7 SUMMARY OF THE 3-PHASED EXPERIMENTS AND FURTHER ANALYSIS

In the three phases of our experiments, we examined our hypothesis step by step: First, we verified that, by optimizing the ReMax objective, a stochastic policy emerges as the optimal policy in a simple bandit problem (Sec. 3). Secondly, we confirmed that stochastic exploration emerges even in a deterministic MDP (Sec. 5). Finally, we demonstrated that this result is consistent in realistic scenarios with neural network function approximation (Sec. 6). In this section, we complement our experiments by showing ablation studies on two components of ReMax: maximization and resetting. Also, we address the overestimation problem in stochastic environments and propose a method to relieve it.

**ReMax objective ablation study.** To demonstrate that not only the resetting but also the maximization in the ReMax return is important for the emergence of a stochastic policy, we change the *maximum* operator in the objective to the *average* operator and compare the performance in the *biased maze* problem (Fig. 7 A). We see that when we use the *average*, the policy becomes deterministic faster and results in poor performance. In App. C.2 we show an extended ablation study, as well as a variant with resetting but while using the standard RL objective.

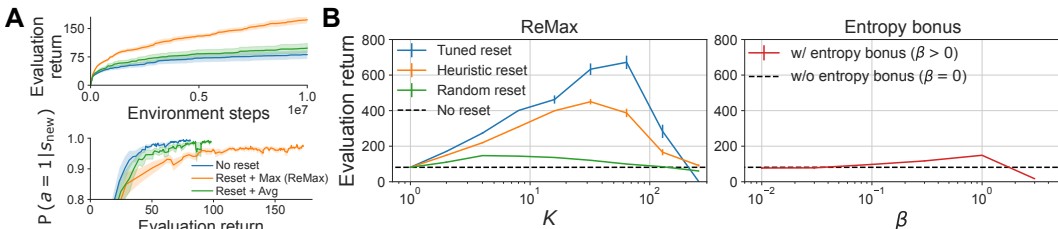

**Figure 7:** Ablation study in *biased maze*. **(A)** Comparison with the *average* variant of ReMax ($K = 2$). **(B)** ReMax performance with different reset policies *(Left)* and performance of REINFORCE with an entropy bonus *(Right)*. The $K$ and $\beta$ are the hyperparameters for exploration. Black dashed lines indicate the performance of the standard REINFORCE without resetting or an entropy bonus.

**Reset policy ablation study.** We also examined the effect of the reset policy on the performance in the *biased maze* task. As the reset policy used in the previous experiments is *well-tuned* using the information of the maze structure, we prepared two other reset policies that utilize no environment information: *random reset* and *heuristic reset*. The *random reset* simply chooses where to reset randomly from the preceding states in the trajectory. The *heuristic reset* returns to the state $s$ where $\pi_\theta(a|s)$ is the smallest among the previously visited states in the preceding trajectory when the agent reaches the terminal state. Note that these two reset policies have only one hyperparameter $K$ and are not tuned using the information of the maze environment. Fig. 7 (B) shows the performance of ReMax REINFORCE with these reset policies. Also, the results of REINFORCE using an entropy bonus are shown for comparison. The *tuned reset* is the same reset policy as in the previous experiments ($m$-step reset with $m = 2$). We found that the choice of reset policy is critical: *tuned reset* performed the best and *heuristic reset* achieved significantly better performance than *random reset*. However, we also found that even with a poor reset policy like the *random reset*, it achieves a comparable

performance to that of the entropy bonus. The performance gap between the *heuristic reset* and the *tuned reset* may be filled by a sophisticated reset policy, which we leave for future work.

**Overestimation in stochastic environments.** The maze environments we studied are *deterministic*. Here we discuss a problem that may arise when optimizing the ReMax objective in *stochastic* environments. We consider another two-armed bandit problem: Taking $a = 0$ gives a reward $r_0 = 1$ deterministically, and $a = 1$ gives a random reward $r_1$, which follows the uniform distribution on $[-10, 10]$. In this bandit problem, the optimal policy in the RL objective is deterministically choosing $a = 0$

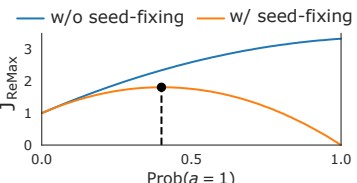

**Figure 8: Overestimation example.**

as $r_0 = 1 > 0 = \mathbb{E}_{r_1}[r_1]$. However, the optimal policy in ReMax is $a = 1$. The blue line in Fig. 8 shows $J_{\text{ReMax}}$ with $K = 2$. In this example, the ReMax objective *overestimates* the action with a high-variance reward because it may randomly achieve a higher reward when playing an arm a second time. To relieve this problem, we propose a simple *seed-fixing* trick: inside an episode, the random seed is frozen. Thus, taking the same actions in the same state would always result in the same state transition. This simple trick can prevent the agent from repeating the action with a high-variance reward as it always gives the same result inside each episode. The orange line in Fig. 8 shows $J_{\text{ReMax}}$ with this trick. Now the greedy action with the optimal policy in ReMax is $a = 0$. Note that the bandit example described in Sec. 3 can be regarded as a problem to which this trick is applied.

## 8 PRACTICAL ALGORITHM: REMAX A2C

We saw that ReMax promotes exploration in the classical REINFORCE algorithm. Here, we demonstrate the feasibility of applying ReMax to promote exploration in modern PG methods. In particular, we apply ReMax to A2C, a synchronous version of the A3C algorithm (Mnih et al., 2016).

**Truncated rollouts with resetting.** Instead of performing actions for a full episode of $T$ time-steps, then updating the policy, A2C updates the policy many times during an episode, using truncated rollouts of length $H < T$. A2C estimates the PG on a batch of such fixed-length truncated trajectories. Using truncated rollouts improves the speed of learning by increasing the frequency of updates, and by reducing the PG variance. ReMax A2C analogously uses a batch of truncated trajectory trees with fixed tree sizes. We describe the detailed rollout procedure in App. F.

**Advantage estimation.** Given a truncated rollout trajectory $\tau$, A2C uses an advantage estimator $\hat{A}_t(\tau) := \sum_{h=t}^{H-1} \gamma^{h-t} r_h + V_\theta(s_H) - V_\theta(s_t)$ composed of the $n$-step future return and a value function baseline, where $\gamma$ is a discount factor, $H$ is the truncated last time-step in $\tau$, and $V_\theta$ is a parameterized value function. Note that if $s_H$ happens to be a terminal state, $V_\theta(s_H)$ is set to zero. As an analogy of $\hat{A}_t$, given a truncated trajectory tree $\mathcal{T}$ by Algorithm 1, we define $\hat{A}_t^{\text{ReMax}}$ using the *best $n$-step future return* and a value function baseline:

$$\hat{A}_t^{\text{ReMax}}(\mathcal{T}) := \max_{k \in \{k | s_t \in \tau^{(k)}\}} \left\{ \sum_{i=i_{k,t}}^{I_k - 1} \gamma^{i - i_{k,t}} r_i^{(k)} + V_\theta(s_{I_k}^{(k)}) \right\} - V_\theta(s_t), \tag{8}$$

where $s_i^{(k)}$ indicates the $i$-th node on the $k$-th trajectory path (from the root node to the leaf node), $I_k$ is the time-step index of the leaf node, and $i_{k,t}$ is the time-step index of state $s_t$ on the $k$-th path satisfying $s_{i_{k,t}}^{(k)} = s_t$ for $k \in \{k | s_t \in \tau^{(k)}\}$. Note that this estimator ignores the rewards before the state $s_t$, while they should be included if we wish to guarantee unbiasedness. However, based on Thm. 1 we can ignore all rewards received before the truncated rollout, and empirically we found that ignoring the rewards from the start of the truncated rollout performed well. In the next section, we empirically verify that ReMax A2C, a PG method with this estimator, also promotes exploration.

### 8.1 EVALUATION ON MINATAR ENVIRONMENTS

Finally, we evaluate our ReMax A2C in the MinAtar environments (Young & Tian, 2019), which include five simplified versions of ALE games with image observations (Bellemare et al., 2013): Asterix, Breakout, Freeway, Seaquest, and SpaceInvaders. Ceron & Castro (2021) reported that algorithmic improvements on MinAtar transferred to the full Atari, thus allowing for more inclusive

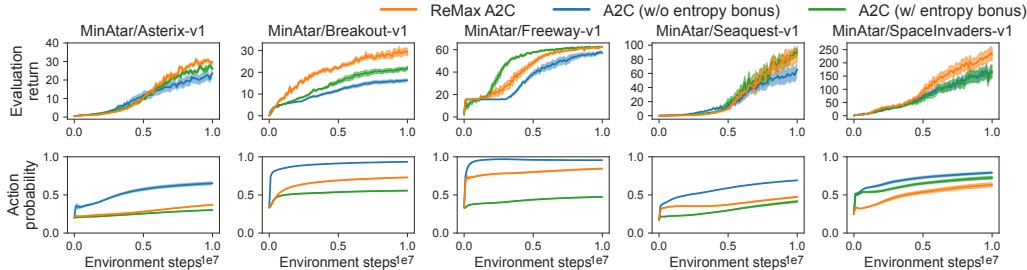

**Figure 9: MinAtar results. (A)** Average evaluation return for 10 runs. **(B)** Average probability of the selected action during the evaluation. The shaded area indicates the standard error.

RL research as it becomes possible to test new methods using fewer computational resources. We compared the performance of our ReMax A2C and the standard A2C (*without* entropy bonus). We also report the performance of standard A2C (*with* entropy bonus) for comparison. Moreover, in App. G we conducted an ablation study on the maximization in ReMax.

**Setup.** We employ the same convolutional neural network architecture as Young & Tian (2019). There are 64 parallel rollout workers, and the rollout length is 32. We use the Adam optimizer (Kingma & Ba, 2015). The reset policy $\pi_\mathcal{X}$ is the *heuristic reset* (see Sec. 7). Whether to reset is decided by $\eta(y = 1|s) = 1$ if $s$ is terminal, and a constant small hyperparameter $p_{reset}$ otherwise. Hyperparameters were grid-searched using five validation runs, which use different random seeds from the test runs. See App. G for the details of the network architecture and hyperparameter selection.

**Results and discussion.** The experiments confirm that ReMax promoted more exploratory policies than A2C (w/o entropy bonus) and lead to better performance in all games (Fig. 9), despite not including any explicit exploration bonus. We note that in the Freeway environment, the episodes always continue for 2500 steps without any early termination by reaching a terminal state, yet our proposal of randomly resetting in non-terminal states with a small probability was sufficient to promote exploration. These results confirm that it is not intractable to construct reset policies that allow taking advantage of ReMax to promote exploration in modern PG algorithms, and even simple heuristics may be sufficient. Comparing A2C (w/ entropy bonus) to ReMax A2C, there was no clear winner. Our main objective in this work is to propose ReMax as a competing or complimentary approach to promote stochastic exploration. As entropy bonuses have been researched for a long time, we believe it is promising that our new method, ReMax, shows competitive performance.

## 9 CONCLUSIONS, LIMITATIONS AND FUTURE WORK

We studied the hypothesis of whether optimizing the ReMax objective results in an exploratory policy by greedily maximizing rewards, without an explicit exploration bonus. The results were consistent with our hypothesis in three phases of experiments: randomized bandit (Sec. 3), the deterministic *biased maze* (Sec. 5), and a maze with visual observations (Sec.6). Moreover, our ablation studies showed that not only the *resetting*, but also the *maximization* part of ReMax was necessary for the emergence of exploration and the improved performance. Finally, we showed that ReMax can be combined with the A2C algorithm leading to a practical scalable RL algorithm based on our method.

The type of promoted exploration is stochastic action selection similar to exploration from an entropy bonus. Thus, it has the same limitations as competing stochastic exploration methods. For example, dense reward signals may be required. Moreover, while the performance was competitive with the entropy-based approach, it did not convincingly improve over this competing method. Lastly, in the present work, we have not explored the full potential of the ReMax objective, and only tested simple reset policies, as our focus was on establishing that any exploration would emerge.

There are many avenues of research along which one may build on the ReMax objective, e.g., advanced resetting strategies. Resetting based on search algorithms such as MCTS, may enhance the exploration further. In addition, we may also train the reset policy by optimizing the RL (or probably the ReMax) objective to discover better exploration strategies depending on the task. We believe that many new directions of research on exploration in RL may emerge from the ReMax objective.

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

# A ADDITIONAL RELATED WORK

This section describes a short survey of exploration in RL.

**Optimism in the Face of Uncertainty (OFU).** Arguably, the most famous exploration strategy is OFU. Exploration methods based on OFU mainly fall in two categories: confidence-based (Strehl & Littman, 2005; Jaksch et al., 2010; Dann & Brunskill, 2015) and bonus-based methods (Strehl et al., 2006; Strehl & Littman, 2008; Azar et al., 2017; Jin et al., 2018). While they have strong theoretical guarantees, they do not directly extend to the deep RL setting, since visitation counts need to be stored. Bellemare et al. (2016) generalize the notion of visitation counts and enable OFU in the deep RL setting.

In contrast to OFU, ReMax uses no explicit mechanism to encourage exploration. Instead, exploration emerges through resetting and seeking the best trajectory.

**Intrinsic Motivation (IM).** An alternative approach that scales well to the deep RL setting is IM-based methods, which are broadly categorized to three types: prediction-error-based, information-gain-based, and novelty-based methods.

Prediction-error-based methods construct a state-transition dynamics model and encourage the agent to visit states (or state-action pairs) whose next state is unpredictable by the model (Stadie et al., 2015; Pathak et al., 2017). This idea dates back to Schmidhuber (1991b) and Thrun & Möller (1991).

Prediction-error-based methods may unnecessarily prefer states at which next states are unpredictable (due to e.g., noise) (Schmidhuber, 1991a). Instead, information-gain-based methods encourage the agent to visit states (or state-action pairs) at which the agent can gain information to refine the model (Schmidhuber, 2010; Sun et al., 2011; Houthooft et al., 2016).

Novelty-based methods literally encourages the agent to visit "novel" states (or state-action pairs). There are different ways to measure the novelty, such as pseudo-count (Bellemare et al., 2016), estimated probability of a state to be contained in a reply buffer (Fu et al., 2017), reachability (Savinov et al., 2019), and intra-episode diversity of states (Badia et al., 2020). Tang et al. (2017) employ a direct approach for the counts to discretize the state space and use hash.

These methods typically require the estimation of some probability density model, such as state-transition dynamics and state visitation frequency. In contrast, our method does not require the estimation of any additional quantity.

RND (Burda et al., 2019) and the E-value (Fox et al., 2018) are interesting exceptions of novelty-based methods that do not require the estimation of a probability density model. It would be an interesting future direction to combine our method with them to determine a state to reset to.

**Entropy Maximization.** Exploration methods based on entropy maximization are often used with policy gradient and actor-critic methods (Williams & Peng, 1991; Mnih et al., 2016; Espeholt et al., 2018). SAC is an example of such methods, and it is a popular deep RL algorithm for continuous control (Haarnoja et al., 2018).

While these approaches aim at increasing the action entropy, increasing the state entropy might be more reasonable from the exploration perspective (Pitis et al., 2020; Mutti et al., 2021) because the action entropy promotes only undirected exploration. Baram et al. (2021) propose to maximize a transition entropy, which is the entropy of a next state conditioned on a current state, as a proxy for the state entropy.

In contrast to these methods, exploration emerges through resetting and seeking the best trajectory in ReMax, as noted before.

**Tree-search.** MCTS (Coulom, 2006) and its variant UCT (Kocsis & Szepesvári, 2006) are popular tree-search algorithms. ExIt (Anthony et al., 2017) combines UCT and deep learning. It uses tree search to find an improved policy, and neural networks imitate it and estimate its value. This approach has been shown to work in complex two-player games (Silver et al., 2017) too. Building on ExIt, Anthony et al. (2019) propose to express a simulation policy by a neural network and train it by the policy gradient method, which removes the necessity of state (and state-action pair) counts in UCT. Temporal-difference search (Silver, 2009) is based on a similar idea, but its simulation policy is the

$\varepsilon$-greedy policy with respect to a Q-function, and the Q-function is updated during the search based on temporal-difference learning.

Our algorithm can be understood as a kind of the tree-search algorithm, wherein it learns a policy maximizing the ReMax objective, and goes back some steps when a leaf node (terminal state) is reached.

**Resetting.** The closest to our method is probably Go-Explore (Ecoffet et al., 2021). It essentially counts the number of visitation to each observed state, resets to a less visited state, and then starts an exploration from it. Counting requires the discretization of the state space, and Go-Explore requires some domain knowledge in general for a reasonable discretization.

In contrast, our method does not require visitation counts. Instead, exploration automatically emerges through the maximization of the ReMax objective with the policy gradient method. Nonetheless, learning when to reset and to which state in our method is an important and interesting direction. Using counts like Go-Explore is one way, but using RND or E-values explained above may be another.

**Change of a Start-state Distribution.** An idea different from but related to resetting is the change of a start-state distribution. It changes the start-state distribution from MDP's initial-state distribution to one that generates states nearby goal states (Florensa et al., 2017; McAleer et al., 2019), and human-expert data distribution (Hosu & Rebedea, 2016; Peng et al., 2018). As in resetting, restart modifies the data distribution for training, whose importance is backed up by some theoretical work (Kakade & Langford, 2002; Munos, 2003; 2005). However, the start-state distribution must be chosen appropriately and often requires side information like expert data or limited task setting such as goal-conditioned MDPs. Therefore, we did not compare our algorithm against those methods.

## B ReMax objective in bandit problem

We can analytically compute the ReMax objective in the bandit problem:

$$J_{\text{ReMax}}^{(K)}(\pi) = 0.25\left(1 - p^K\right) + 0.75\left(1 - (1-p)^K\right), \tag{9}$$

where $p := \pi(a = 1)$. The first term represents the probability that the correct arm is 0, and the agent plays it at least once. The second term is that of the action $a = 1$. Fig. 1 (A) compares $J_{\text{RL}}$ and

$$J_{\text{ReMax}}^{(2)}(\pi) = -p^2 + 1.5p + 0.25. \tag{10}$$

We can confirm that $p = 0.75$ (shown as the black dotted line in Fig. 1 A) maximizes the ReMax objective, and the optimal policy under the ReMax objective is exploratory.

## C Biased maze ablation study

We ran several additional experiments in the *biased maze* problem (Sec. 5) to analyze the behavior of ReMax REINFORCE in detail.

### C.1 Bias term effect

In our experiment, the standard REINFORCE got stuck in a deterministic suboptimal solution (Fig. 4). This result may be confusing as the policy $\pi(a = 1|s_i) = \sigma(w_i + c)$ has a weight parameter $w_i$ for each state. Learning an appropriate $w_i$ for each state must be enough to solve the maze. It happens because the bias term $c$ allows the policy to control the "prior" behavior at unseen states. Note that $c$ does not depend on the state index $i$, and is expected to learn the global maze bias. Fig. 10 shows the results for the standard REINFORCE with and without bias term $c$. The learning rate was 0.03. Without $c$, the algorithm does not get stuck in a deterministic suboptimal solution because it cannot control the behavior at unseen states. As $w_i$ is initialized to zero, the probability to select action one at a new state is always 0.5. This initial distribution for unseen states is good enough to keep going on the maze. However, our interest here is the algorithm behavior when the policy can control its exploratory behavior, and thus we added the bias term $c$ in our experiment. In this case, an algorithm may fail to solve the maze if it has a poor exploration mechanism, as the standard REINFORCE actually did. This is also the case when the policy uses a neural network as function approximator.

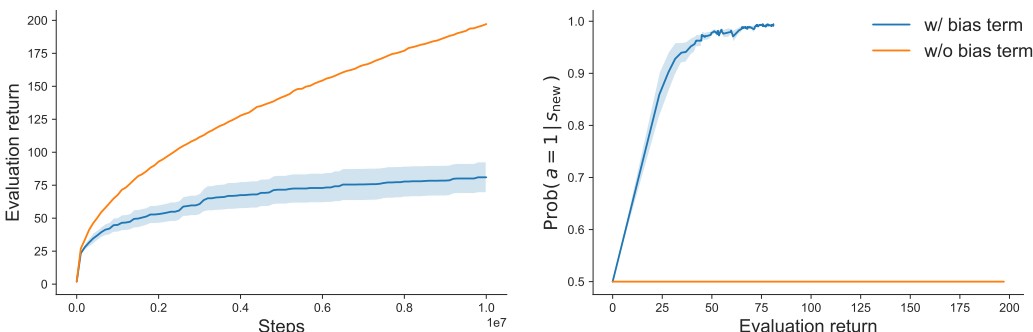

**Figure 10: REINFORCE performance with/without bias term** *c*. Average evaluation return of ten runs *(Left)* and average probability to choose action one at a new state *(Right)*.

In Sec. 7, we compared the ReMax objective and its *average* variant (instead of *maximization*). Here, we show the results of another variant and the results with different $K$. We consider two variants:

- *average*: we replaced the maximizing operation in ReMax objective by averaging operation.
- *separated*: we regard a trajectory tree as separated trajectories, and apply the standard RL objective and REINFORCE algorithm to them.

See Fig. 11 for the results with the *average* variant and Fig. 12 for the *separated* variant. The learning rate was 0.03. Although both variants benefit from sampling with *well-tuned* resetting, they are not as exploratory as the ReMax objective, and we see that their performance are worse than the ReMax objective.

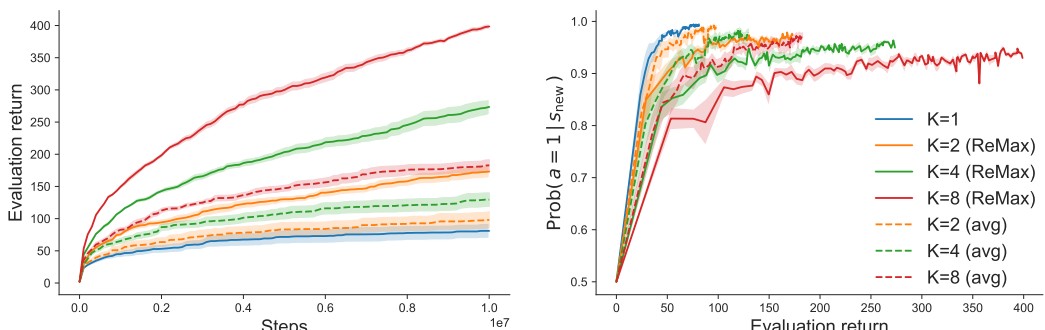

**Figure 11: Results of *average* variant.** Evaluation return *(Left)* and probability to select action one at a new state *(Right)*. Dotted lines indicate the results of *average* variant.

# D PROOF OF THE RESET POLICY GRADIENT THEOREM

**Theorem 1** (Reset Policy Gradient Theorem). *Denote $\tau^*$ is the optimal trajectory in the tree, $\mathcal{T}$, so that $\mathcal{R}_{\mathrm{ReMax}}(\mathcal{T}) = \mathcal{R}(\tau^*) = \sum_{(s,a) \in \tau^*} r(s, a)$. Moreover, denote $\tau_{\mathrm{fixed}}(s_t)$ is a subsequence $\tau_{\mathrm{fixed}}(s_t) \in \tau^*$, starting at $s_0$ and ending at $s_h$, $h < t$, s.t. for all possible trajectories with $k \geq t$, and all $s \in \tau_{\mathrm{fixed}}(s_t)$ we have $s \notin \mathcal{X}(s_k)$, where $\mathcal{X}(s_k)$ is the set of admissible states to reset to in state $s_k$. In other words, $\tau_{\mathrm{fixed}}(s_t)$ is the set of states in the optimal trajectory to which it is impossible to reset to when starting at state $s_t$, at any point following time-step, $t$. Then we have*

$$\nabla_\theta \mathbb{E}\left[\mathcal{R}_{\mathrm{ReMax}}(\mathcal{T})\right] = \mathbb{E}\left[\sum_{t|u_t=a_t \in \mathcal{A}} \nabla_\theta \log \pi_\theta(a_t|s_t)\big(\mathcal{R}_{\mathrm{ReMax}}(\mathcal{T}) - \mathcal{R}\left(\tau_{\mathrm{fixed}}(s_t)\right) - b_t\big)\right] \quad (11)$$

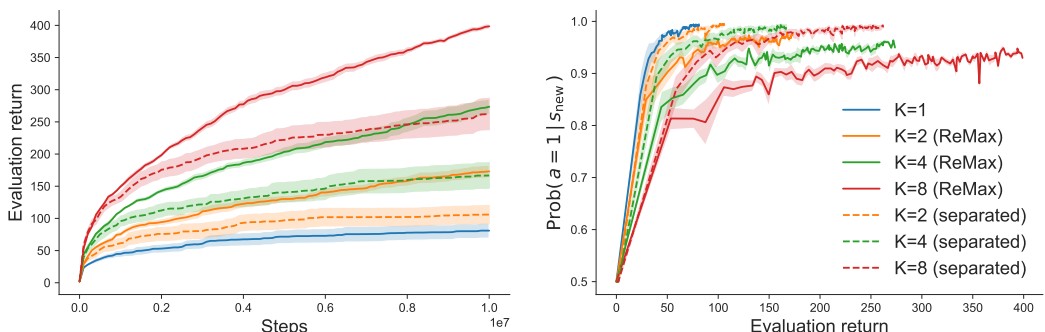

**Figure 12: Results of *separated* variant.** Evaluation return *(Left)* and probability to select action one at a new state *(Right)*. Dotted lines indicate the results of *separated* variant.

*Proof.* Starting from the unbiasedness of Eq. 5, we have

$$
\nabla_\theta \mathbb{E}\left[\mathcal{R}_{\text{ReMax}}\left(\mathcal{T}\right)\right] = \mathbb{E}\left[\sum_{t|u_t=a_t\in\mathcal{A}} \nabla_\theta \log \pi_\theta(a_t|s_t)\big(\mathcal{R}_{\text{ReMax}}(\mathcal{T}) - b_t\big)\right]
$$

$$
= \mathbb{E}\left[\sum_{t|u_t=a_t\in\mathcal{A}} \nabla_\theta \log \pi_\theta(a_t|s_t)\big(\mathcal{R}\left(\tau^*\backslash\tau_{\text{fixed}}(s_t)\right) + \mathcal{R}\left(\tau_{\text{fixed}}(s_t)\right) - b_t\big)\right]
$$

$$
= \mathbb{E}\left[\sum_{t|u_t=a_t\in\mathcal{A}} \nabla_\theta \log \pi_\theta(a_t|s_t)\big(\mathcal{R}\left(\tau^*\backslash\tau_{\text{fixed}}(s_t)\right) - b_t\big)\right]
$$

$$
+ \mathbb{E}\left[\sum_{t|u_t=a_t\in\mathcal{A}} \nabla_\theta \log \pi_\theta(a_t|s_t)\big(\mathcal{R}\left(\tau_{\text{fixed}}(s_t)\right)\big)\right]
$$

$$(12)$$

It remains to be shown that $\mathbb{E}\left[\sum_{t|u_t=a_t\in\mathcal{A}} \nabla_\theta \log \pi_\theta(a_t|s_t)\big(\mathcal{R}\left(\tau_{\text{fixed}}(s_t)\right)\big)\right] = 0$. It is well known that $\mathbb{E}_{\pi_\theta}\left[\nabla_\theta \log \pi_\theta(a)Y\right] = 0$, for a random variable $Y$ statistically independent to $a$, because $\mathbb{E}_{\pi_\theta}\left[\nabla_\theta \log \pi_\theta(a)Y\right] = \mathbb{E}_{\pi_\theta}\left[\nabla_\theta \log \pi_\theta(a)\right]\mathbb{E}_{\pi_\theta}\left[Y\right]$, and $\mathbb{E}_{\pi_\theta}\left[\nabla_\theta \log \pi_\theta(a)\right] = \nabla_\theta\mathbb{E}_{\pi_\theta}\left[1\right] = 0$. Finally, note that $\mathcal{R}\left(\tau_{\text{fixed}}(s_t)\right)$ will not change, irrespective of the choice of $a_t$ based on the definition of $\tau_{\text{fixed}}(s_t)$; hence, it is statistically independent to $a_t$, and the expectation is 0. $\qquad\square$

## E    HYPERPARAMETER SELECTION IN MNIST MAZE PROBLEM

In the MNIST maze experiments (Sec. 6), we aimed to examine the behavior of ReMax REINFORCE when the policy appropriately utilizes the information from the image input to solve the maze. Therefore, we searched the hyperparameters, the learning rate and the image input scale $d$ so that the policy can solve the maze well when the image flip probability is temporarily set to 0, i.e. when the agent can solve the maze perfectly if it blindly believes the MNIST image hint. We performed ten validation runs with different seeds than the test runs and grid-searched the hyperparameters. The search space was $\{0.0003, 0.001, 0.003, 0.01, 0.03\}$ for the learning rate and $\{1.0, 0.1, 0.01\}$ for $d$. Fig. 13 shows the learning curve of top five hyperparameter sets in validation runs. Based on this experiment, we used 0.003 for the learning rate and 0.01 for $d$. Note that, with those hyperparameters, the policy quickly learned to reach the maximum length of maze (1000).

## F    TRUNCATED ROLLOUTS WITH RESETTING

The rollout procedure of each rollout worker is described in Algorithm 1. Each rollout function call receives the state $s_{\text{start}}$, from which the rollout starts and returns a fixed-size trajectory tree and the start state $s_{\text{start}}$ for the next rollout. Note that the initial $s_{\text{start}}$ is sampled from $\rho_0$. Instead of setting

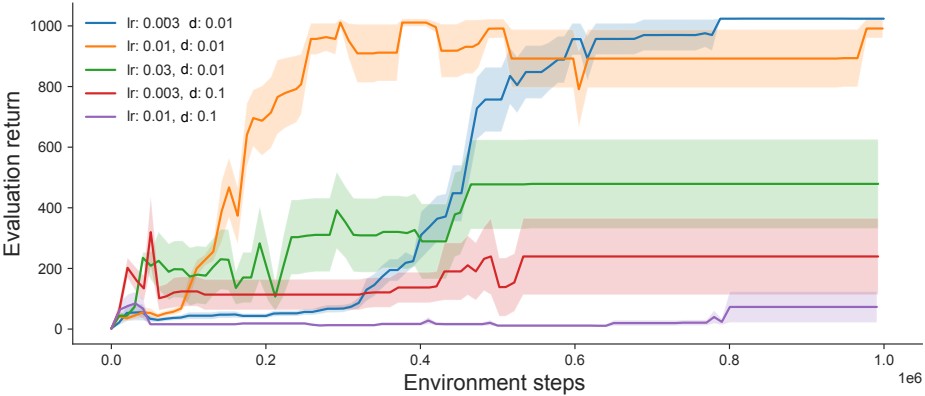

**Figure 13: Top five hyperparameters in validation runs.**

a fixed number of resets $K$ as previously, the resetting will continue for as many times as necessary until the fixed tree size is reached. Moreover, as there is the possibility of no resets occurring if the agent does not encounter a terminal state, we introduced a hyperparameter $p_{\text{reset}}$ that determines the probability of resetting at non-terminal states: $\eta(y = 1|s) = 1$ if $s$ is a terminal state and $p_{\text{reset}}$ otherwise.

---

**Algorithm 1** Rollout function with reset

---

**Input:** $s_{\text{start}}$
  Initialize $t = 0$, $s = s_{\text{start}}$, $\mathcal{X} = \{s\}$, and $\mathcal{T} = \{s\}$
  **while** $t < H$ **do**
    Determine *whether to reset or not* $y \sim \eta(\cdot|s)$
    **if** $y == 1$ **then**
      Add $s$ to leaf nodes
      Determine *where to reset* $x \sim \pi_{\mathcal{X}}(\cdot|s)$
      Update $s \leftarrow s_{\text{Re}}(x)$                                             ▷ Reset to $s_{\text{Re}}(x)$
      **continue**
    **else**
      Sample $a \sim \pi_{\theta}(\cdot|s)$ and $s' \sim P(\cdot|s, a)$
      Update $\mathcal{T} \leftarrow \mathcal{T} \cup \{a, s'\}$ and $\mathcal{X} \leftarrow \mathcal{X} \cup \{s'\}$
      Update $t \leftarrow t + 1$ and $s \leftarrow s'$
    **end if**
  **end while**
  Add $s$ to leaf nodes
  Sample $s_{\text{start}}$ randomly from leaf node states
  **if** $s_{\text{start}}$ is terminal **then** $s_{\text{start}} \sim \rho_0(\cdot)$
  **return** $\mathcal{T}, s_{\text{start}}$

---

## G  MINATAR EXPERIMENT DETAILS

**Neural network architecture.** We employed the same neural network architecture in the original MinAtar study (Young & Tian, 2019) to ensure a reasonable baseline algorithm performance. The architecture consists of one convolutional layer, one linear hidden layer, and one fully connected output player. The convolutional layer has 16 output channels from $3 \times 3$ kernel with stride size 1. Both hidden layer and output layer have 128 units. The SiLU and dSiLU activation functions (Elfwing et al., 2018) are used after the convolutional layer and hidden layer, respectively. The policy and value function share the network.

**Ablation study.** To confirm that the ReMax objective enhances the exploration, we compared the ReMax A2C with a variant that uses resetting but not the ReMax objective as we did in the biased maze experiments (Sec. C.2). We found that the separated variant, which regards a trajectory tree as

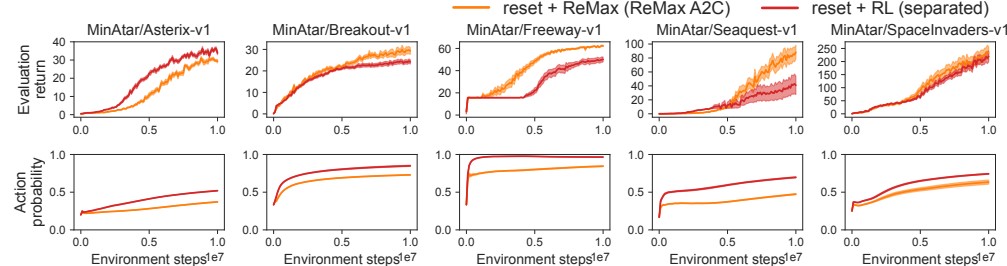

**Figure 14: MinAtar ablation.** **(A)** Average evaluation return for 10 runs. **(B)** Average probability of the selected action during the evaluation. The shaded area indicates the standard error.

a set of separated trajectories and uses RL objective, becomes more deterministic in all games and performs poorly on 4/5 MinAtar games compared to the ReMax A2C.

**Hyperparameter selection with validation runs.** In the MinAtar experiments, we used learning rate $\alpha = 0.003$ for all experiments, which performed best with all algorithms in the preliminary experiments. We determined the other hyperparameters using the results of validation runs. The hyperparameter for A2C (w/ entropy bonus) is $\beta$ for controlling the scale of entropy bonus term. In ReMax A2C, the hyperparameter is $p_{\text{reset}}$, probability that reset happens at a non-terminal state. The *separated* variant of ReMax A2C also has $p_{\text{reset}}$ as the hyperparameter. In each of the MinAtar games, we selected the hyperparameters which maximized the average evaluation return of five validation runs. The results of validation runs are shown in Table 1, Table 2, and Table 3.

**Table 1:** Validation return of A2C (w/ entropy bonus)

| $\beta$ | Asterix | Breakout | Freeway | Seaquest | SpaceInvaders |
|---|---|---|---|---|---|
| 0.01 | 23.634 | 18.609 | 61.859 | **99.253** | **197.897** |
| 0.03 | **26.744** | 20.997 | **62.766** | 54.888 | 185.309 |
| 0.1 | 17.291 | **24.331** | 46.463 | 4.044 | 183.844 |
| 0.3 | 6.194 | 11.813 | 19.375 | 1.209 | 41.784 |

**Table 2:** Validation return of ReMax A2C

| $p_{\text{reset}}$ | Asterix | Breakout | Freeway | Seaquest | SpaceInvaders |
|---|---|---|---|---|---|
| 0.0 | **36.353** | 25.544 | 43.081 | 94.934 | 238.434 |
| 0.02 | 35.659 | **25.947** | 62.284 | **103.978** | **257.588** |
| 0.05 | 32.403 | 24.744 | **62.409** | 93.475 | 187.694 |

**Table 3:** Validation return of *separated* variant

| $p_{\text{reset}}$ | Asterix | Breakout | Freeway | Seaquest | SpaceInvaders |
|---|---|---|---|---|---|
| 0.0 | 35.675 | 21.65 | 46.722 | 70.553 | **234.091** |
| 0.02 | 34.616 | 23.788 | 57.188 | 28.05 | 202.994 |
| 0.05 | **39.041** | **26.809** | **58.472** | **74.928** | 210.713 |

