# OpenReview forum: "Emergence of Exploration in Policy Gradient Reinforcement Learning via Resetting"
_ICLR.cc/2023/Conference — Submitted to ICLR 2023_

### Official Review · Reviewer_AvuR · 2022-10-23

**Confidence:** 4
**Clarity, Quality, Novelty And Reproducibility:** See above
**Correctness:** 1
**Technical Novelty And Significance:** 1
**Empirical Novelty And Significance:** 1
**Recommendation:** 1

**Strength And Weaknesses:**

Major Weaknesses:
1. The paper fails to motivate the need for their proposed objective function when there are several existing exploration based objectives and algorithms that achieve state-of-the-art results on many environments.
2. The performance of the policy obtained with ReMax objective is not shown against the actual reward of the environment we wanted to optimize. One could also define their objective as maximizing the entropy of actions and show that taking random actions at every time step maximizes their defined objective. It misses the whole point of optimizing for the actual objective and I fail to see the utility of these objectives and corresponding algorithms.
3. There aren't any experiments comparing ReMax with other exploration based methods
4. "As our main focus in this study is the emergence of exploration from our ReMax objective, we only employ simple deterministic rule-based policies for πX and η". This seems like exploration did not 'emerge' but is being imposed by the epsilon value which determines whether to explore or not given a particular state.
5. A stochastic policy is the one where each action output has a probability associated with it. One could define a policy network in such a manner. I do not understand how an objective function can encourage stochasticity. Relatedly, what is a "stochastic exploration"?
6. The proposed algorithm isn't novel. As mentioned in the paper, this idea has been explored in Go-Explore. Moreover, it seems like a very inferior version of MCTS (which has a more principled way of exploring different future states without arbitrarily resetting to previous states)

Minor weaknesses:
1. It mentions entropy bonus based objectives but fails to mention several other algorithms like count-based exploration (Tang, 2016) or intrinsic rewards (Zheng, 2018)
2. Insufficient detail about the maze environment and results. How long is the maze? What is 'maximum sufficient fixed length'? Why does it need 10M training steps for such a seemingly simple environment?
3. Figure-4b is unclear. Why does it cause such a huge variance with changing learning rate? Is the experiment performed with the learning rates 1e-3, 1e-2, 1e-1 or other intermediate learning rates too? Why is this comparison relevant? Aren't we just interested in the performance with best learning rate? Are other hyper parameters tuned too?

Questions:
1. When the agent resets to a previous state, does the time_steps also reset to their old value? What prevents it from going in an infinite-loop or never-ending episodes?
2. What is the rationale for choosing policy gradient methods and not value-based methods?

**Summary Of The Paper:**

It proposes a novel objective function (ReMax) and a novel algorithm of resetting to previous states for encouraging implicit 'stochastic exploration' policies.

**Summary Of The Review:**

Based on the major weaknesses listed above, I recommend to reject this paper.

---

> ### Author Response · Authors · 2022-11-07
> **Response [2/2]**
>
> >Insufficient detail about the maze environment and results. How long is the maze? What is 'maximum sufficient fixed length'? Why does it need 10M training steps for such a seemingly simple environment?
>
> The maze is 1000 steps long (sufficient length just meant that we set it large enough that the agent does not reach the end during learning and can continuously improve). Regarding the learning speed, if there were no bias term in the policy, we could set the learning rate to be large, and the learning would be much faster. However, due to the bias term, setting the learning rate large causes the policy to become deterministic at newly visited states and the learning to stop. To overcome this, the learning rate has to be set sufficiently small so that the learning is stable.
>
> >Figure-4b is unclear. Why does it cause such a huge variance with changing learning rate? Is the experiment performed with the learning rates 1e-3, 1e-2, 1e-1 or other intermediate learning rates too? Why is this comparison relevant? Aren't we just interested in the performance with best learning rate? Are other hyper parameters tuned too?
>
> As this is a simple task, increasing the learning rate makes it learn faster. However, the learning becomes unstable. We opted for showing the performance curves with a stable learning rate, but also plotted the final performance for other learning rates (our method improved over the baseline for all learning rates). The experiment was performed with the learning rates 1e-3, 3e-3, 1e-2, 3e-2, 1e-1. Showing that the method improves across all learning rates is a much stronger result than showing only the improvement at the best learning rate. The robustness to the hyperparameters is also an important property of algorithms. There are no other hyperparameters.
>
> **Questions:**
> > When the agent resets to a previous state, does the time_steps also reset to their old value? What prevents it from going in an infinite-loop or never-ending episodes?
>
> In the maze tasks, the maximum number of resets was set by the parameter $K-1$ ($K$ is the number of leaf nodes, so $K-1$ is the number of resets). In the MinAtar tasks, the trajectory tree size was fixed to 32.
>
> >What is the rationale for choosing policy gradient methods and not value-based methods?
>
> Unlike traditional RL, the ReMax objective depends on the full trajectory (including past states and rewards). In this case, the Bellman equation does not hold, and it is not clear how to use value-based methods. However, the policy gradient method can easily be used to obtain an unbiased gradient estimator for the ReMax objective.

---

> ### Author Response · Authors · 2022-11-07
> **Response [1/2]**
>
> Thank you for the review.
>
> >motivate the need for their proposed objective function when there are several existing exploration based objectives and algorithms that achieve state-of-the-art results on many environments.
>
> While there exist many SOTA exploration algorithms, exploration is far from solved. In our work we propose a new direction for exploration research, establish the main idea and prove that it leads to exploration. We believe our approach is interesting as exploration arises by greedily maximizing the rewards without any explicit exploration bonus.
>
> >The performance of the policy obtained with ReMax objective is not shown against the actual reward of the environment we wanted to optimize.
>
> This seems like a misunderstanding. The evaluation returns are with respect to the actual RL objective with no resetting.
>
> >There aren't any experiments comparing ReMax with other exploration based methods
>
> It seems the review may have missed the comparison with entropy based methods?
> Please also see our response to Z1SD regarding why there are no comparisons to further algorithms.
>
> >... , we only employ simple deterministic rule-based policies for πX and η". This seems like exploration did not 'emerge' but is being imposed by the epsilon value which determines whether to explore or not given a particular state.
>
> We clarify that these parameters do not determine whether to explore or not but whether to reset or not. Moreover, our ablation studies in section 7 and appendix C.2 showed that resetting alone does not lead to the emergence of a stochastic policy, but the ReMax objective was necessary.
>
> >A stochastic policy is the one where each action output has a probability associated with it. One could define a policy network in such a manner. I do not understand how an objective function can encourage stochasticity. Relatedly, what is a "stochastic exploration"?
>
> We clarify this point. We indeed define the policy network in this manner (a softmax probability over actions). However, the problem with this approach is that merely optimizing the RL objective with such a policy leads to the policy quickly becoming deterministic and the learning to stop (this can be seen in all 3 phases of experiments as well as in the MinAtar experiments). To stop this problem from happening, we need methods to promote the policy to stay stochastic, e.g., adding an entropy bonus. Stochastic exploration means exploration where the actions are chosen with some randomness. We use the term “stochastic exploration” to contrast it with other more directed approaches, such as smart exploration strategies that plan to go to a state that was not visited before.
>
> >The proposed algorithm isn't novel. As mentioned in the paper, this idea has been explored in Go-Explore.
>
> While resetting based approaches have been used before, the ReMax objective is new. We are not aware of a method that achieves a stochastic exploratory policy by directly maximizing the rewards without any explicit exploration bonus such as our method. We believe this is novel.
>
> >It mentions entropy bonus based objectives but fails to mention several other algorithms like count-based exploration (Tang, 2016) or intrinsic rewards (Zheng, 2018)
>
> Please see the extended related work section in appendix A where we discuss both count-based exploration and intrinsic reward based methods.

---

### Official Review · Reviewer_6LaJ · 2022-10-27

**Confidence:** 4
**Correctness:** 2
**Technical Novelty And Significance:** 3
**Empirical Novelty And Significance:** 2
**Recommendation:** 3

**Clarity, Quality, Novelty And Reproducibility:**

## Clarity
The authors did a fair job in describing their approach and the paper results quite clear.

## Quality
The presented results appears to be sound from both the theoretical and experimental sides.

## Novelty
The idea appears to be quite novel, and the similarity with other works are discussed in the introduction. A related work section is provided in the appendix, however, the article may benefit from including a dedicated related work section in the main paper.

## Reproducibility
The presented experiments are reproducible.

**Strength And Weaknesses:**

## Strengths
The paper deals with the interesting topic of exploration, which is of paramount interest in the RL community, since it is deeply connected with the challenging topic of generalization in RL. The main goal of the article is to re-phrase the exploration problem as the necessity for the agent of visiting the same states multiple times. The author claim that this can be achieved by allowing the agent to properly reset the trajectory.

## Weaknesses
However, the evidence the authors provide to prove their thesis does appear completely convincing. First, it is not clear, from a theoretical point of view, how the ReMax objective optimal solutions relate to the standard objective ones. Does the optimal ReMax solution have some bias w.r.t. to the standard one? Second, it is difficult to state whether the authors claim, that the policy obtained with their strategy promote exploration, is true or not. While the strategy seems to be effective in the worked examples, in the MinAtar benchmark the entropy bonus seems to be more or equally effective w.r.t. the expected return obtained, in all the environment a part from Seaquest.
W.r.t. this aspect, it is not clear to me why the authors affirm "Comparing A2C (w/ entropy bonus) to ReMax A2C, there was no clear winner.".
In order to better understand the effectiveness of the approach in promoting exploration, it would be interesting to compare the two strategy also w.r.t. exploration metrics.

Moreover, the paper lacks a in-depth study of the impact of employing different strategy for resetting and selecting the reset state, and heuristics are used instead. In practice, these strategies could have a fundamental role in the development of the desired exploratory behavior, thus, more should be done to investigate how to properly build them.
Finally, the proposed approach has the strong limitation of requiring the availability of a simulator, differently from entropy based approaches which don't have such requirement.


**Summary Of The Paper:**

The paper proposes an alternative objective for Reinforcement Learning called ReMax. The authors claim that an agent optimizing this objective naturally develops an exploratory behavior, differently from standard RL, where exploration typically should be enforced, for instance adding an entropy bonus. The main idea consists in allowing the agent to reset its trajectory to a specific state, instead of explicitly adding exploration.
The authors provide a policy gradient theorem to optimize such objective, and analysed the obtained policies in three worked examples where the algorithm was applied. Finally, the provided strategy is tested the MinAtar benchmark, in terms of performance w.r.t. the standard RL objective.

**Summary Of The Review:**

While the idea is interesting, the provided results look still preliminary, and do not seem to be still strong enough to support the authors hypothesis. It appears that the presented method does not offer a better or equivalent way of inducing exploratory behavior w.r.t. to state-of-the-art techniques, which do not suffer from the limitation of requiring a simulator.

---

> ### Author Response · Authors · 2022-11-07
> **Response**
>
> Thank you for the review.
>
> >The main goal of the article is to re-phrase the exploration problem as the necessity for the agent of visiting the same states multiple times.
>
> We want to clarify a slight but important difference. It is not that visiting the same state is important for exploration. We claim the opposite. Exploration is important only when the agent visits the same state multiple times (for example when the agent tries the same task multiple times and tries to improve). To maximize the reward at future visits to the same state, the agent should explore. Based on this motivation, we proposed the ReMax objective.
>
> >Does the optimal ReMax solution have some bias w.r.t. to the standard one?
>
> In deterministic MDPs, there is no bias (explained below Eq. 3). However, in stochastic MDPs there is a bias (we show a way to lessen this effect in Sec. 7, see figure 8). Note that the competing entropy based approach is biased even for deterministic MDPs.
>
> >Second, it is difficult to state whether the authors claim, that the policy obtained with their strategy promote exploration, is true or not.
>
> While it is true that it is not clear whether the method is better than the entropy based approach, it is clear that the method does promote exploration. The performance is much better than the standard RL objective with no entropy bonus. Moreover, regarding exploration-based metrics, we plotted the probability of action selection during training for the three methods, and these results showed clearly that the standard RL quickly becomes deterministic, while both the entropy-based approach and the ReMax approach had stochastic action selection. As the performance was comparable to the performance of the entropy based approach we claim that it provides an effective solution to the problem of the standard RL objective. Moreover, our approach is novel, and there are several avenues of research along which the approach could be improved (e.g. better reset policies), so we believe it is promising.
>
> > the paper lacks a in-depth study of the impact of employing different strategy for resetting and selecting the reset state
>
> We did include comparisons of the “tuned”, “random” and “heuristic” reset methods in section 7. While we agree that it is a good research direction to study different reset strategies, we believe this is future work (e.g. trained reset policies would be interesting). The objective of the current work is merely to introduce the ReMax objective and establish that it leads to any non-trivial exploration. It is not obvious that ReMax will outperform the standard RL objective or lead to a stochastic policy, and we have provided strong evidence in our paper that it does.

---

### Official Review · Reviewer_Z1SD · 2022-11-02

**Confidence:** 4
**Correctness:** 2
**Technical Novelty And Significance:** 2
**Empirical Novelty And Significance:** 1
**Recommendation:** 1

**Clarity, Quality, Novelty And Reproducibility:**

The paper is clear and easy to follow. Concerns with quality/significance of experiments and novelty are as outlined above.

**Strength And Weaknesses:**

Weaknesses

1. No analysis/comparison of relation to prior exploration approaches, Quality of experimental environments

The main focus of the paper seems to be to establish that optimizing the reMax objective leads to learning a stochastic policy, and that this is desirable because it leads to better exploration performance. However, it is unclear how well the learned stochastic policy stacks up compared to prior exploration approaches, in any of the 3 phases considered (bandits, biased maze and  maze with image inputs). The only comparison to a different exploration approach was for A2C with entropy bonus on the minAtar envs, where the proposed approach gets higher return on only 1 out of 5 environments (Fig 9). There have been numerous exploration algorithms studied in tabular settings, specifically for bandits, with detailed analysis and none of them have been discussed in relation to the experiments done in phase 1. Regarding the maze environment, exploration algorithms have been shown to solve much more complex visual mazes like VizDoom [1]. Again the authors do not compare against or discuss how the proposed approach's exploration strategies would differ. This is further worrying because prior work has used exactly the settings used here (bandits and visual mazes).

2. Significance of the idea

The idea of resetting to a past state to enhance exploration has been studied in Go-explore [2], as the the authors point out themselves. In go-explore, the environment is reset to states that are rarely encountered, either directly or via a goal-reaching policy. This is very similar to the idea in the proposed paper of maximizing return after resetting to an arbitrary past encountered state in the trajectory. The authors don't include any analysis or comparison against go-explore, or any discussion of which settings would the proposed resetting scheme be better. From an algorithmic perspective, it seems Go-explore should be more exploratory since it resets to arbitrary states, while this paper only considers rests to past states in the same trajectory. Furthermore, Go-explore is evaluated on very challenging exploration environments (Montezuma's revenge) as well as challenging simulated robotic manipulation environments.

[1] Pathak, Deepak, et al. "Curiosity-driven exploration by self-supervised prediction." International conference on machine learning.
[2] Ecoffet, Adrien, et al. "Go-explore: a new approach for hard-exploration problems."


**Summary Of The Paper:**

The paper proposes an approach for training exploratory policies, without adding any explicit exploration bonus. They formulate a new objective (ReMax) which allows resetting to an arbitrary prior state (hence generating new sequences), and then maximizing the best return over the set of sequences, and argue that this leads to the emergence of a stochastic policy. The paper is presented with analysis in 3 phases - random bandit arm (simple partially observable env), biased maze (deterministic fully observed env, uses simple model) and maze with image inputs (deterministic env, uses network model) .


**Summary Of The Review:**

I am in favor of rejecting this paper because there is no comparison or discussion of the exploration performance to that of previous methods (except on the minAtar envs where the proposed method performs better on only 1/5 envs). Further, the proposed idea is very similar to that of go-explore (which obtained state of the art results on montezuma's revenge), again without any comparison or analysis of why/when this method should be used instead.

---

> ### Author Response · Authors · 2022-11-07
> **Response**
>
> Thank you for the review.
>
> The review correctly identifies that our main objective is to establish that optimizing ReMax leads to the emergence of a stochastic exploratory policy. We note that this claim is non-trivial. Simply optimizing the standard RL objective leads to the policy quickly becoming deterministic and the learning to stop. As ReMax also directly optimizes the rewards, it is not clear that it will greatly outperform the standard RL method. We believe our results clearly demonstrate that ReMax does lead to exploration, and greatly outperforms RL with no exploration bonus. Moreover the result was competitive with the entropy bonus based method, showing that the method is effective. We compare with the entropy-based method because it is the most closely related competitor.
>
> We give some more explanations regarding why other comparisons are not included:
>
> **Bandits in Phase 1:** We note that phase 1 is a trivial problem added to demonstrate the principle behind why ReMax leads to exploration. We do not evaluate the performance on this bandit task (a 1 state POMDP), and just illustrate that the optimal ReMax policy is stochastic. We believe comparisons with bandit algorithms are out of the scope of the paper, because we proposed an algorithm for RL, and without envisioning for it to be used for bandit tasks in general.
>
> **[1] Curiosity based exploration:** We note that different exploration strategies are not mutually exclusive. For example, if you look at the code of [1] https://github.com/pathak22/noreward-rl/blob/master/src/constants.py they add an entropy bonus with the entropy coefficient into their A3C algorithm. So, the work is actually using a combination of an entropy bonus together with curiosity based exploration (and their method seems certain to perform poorly without this bonus due to the policy collapsing to a deterministic one). Likewise, we could envision an approach combining ReMax with curiosity based exploration. In our paper we have removed such additional components of the learning algorithm, and simply compare entropy and ReMax directly. We believe that testing the comparison of entropy and ReMax in different settings (e.g., when combined together with curiosity based exploration) is non-crucial.
>
> **Go-Explore:** Like with the curiosity based exploration, Go-Explore could also be combined together with ReMax. First we note that the Go-Explore approach does not include the trajectory tree formalism or the maximization of the rewards in the tree, which was shown in our work to be crucial for the emergence of the stochasticity (so our work is novel). We also argue that Go-Explore is a complex algorithm with many components. In the “Go” phase of the algorithm, the agent is reset to a rarely visited state, then in the “Explore” phase the agent selects actions randomly for 100 steps to explore. Selecting the actions randomly is an explicit method of exploration. In their article they write “we believe exploring intelligently (e.g. via a trained policy) would likely improve our results and is an interesting avenue for future work.” regarding replacing the random action selection with a trained policy. This trained policy could be promoted to be stochastic either via an entropy bonus, or, for example, via ReMax. In this way, the benefits of ReMax and Go-Explore can be combined. For this reason, we find that a comparison with Go-Explore is non-crucial, and we compare with the entropy bonus approach that is a more closely related competitor.
>
> In our work we introduced a new direction for exploration in RL. While it is not clear that the method should be preferred over the entropy based approach in its current state, it appears to be a viable competitor. Moreover, there are many directions of future research (e.g., different reset policies) that could further improve the performance. The aim of the current paper was to establish the main idea, and we believe that further large enhancements to the method are not within the scope of the current paper.

---

### Official Review · Reviewer_KQHn · 2022-11-03

**Confidence:** 4
**Correctness:** 2
**Technical Novelty And Significance:** 3
**Empirical Novelty And Significance:** Not applicable
**Recommendation:** 3

**Clarity, Quality, Novelty And Reproducibility:**

The paper is fairly clear and novel. The overall quality is also good, but I think the experiments can be improved/reframed.

**Strength And Weaknesses:**

Strengths
- The paper shows the emergence of stochastic policies when resetting the agent to previous states in the trajectory and using the ReMax return. This is an interesting contribution, and could lead to future research directions in how to develop exploration policies.


Weaknesses
- The motivation of this paper is that directly optimizing for this objective results in a stochastic exploration policy. It would be helpful to see how this policy actually works as an exploration policy used to train another policy that is trying to optimize the original RL objective.
- The paper compares against other methods that don't reset the simulator during training. The comparison is done based on the number of steps taken in the environment. I am not sure this is a valid comparison. Resetting the environment involves hacking the environment. Other work that hack the environment in such a fashion do so to study a phenomenon that emerges (Go-explore) which is fine. If you're comparing to other methods that don't hack the environment, you need to make allowances to account for that, since doing so is not really possible for most problems of interest. For example, if the agent resets to state A, it automatically skips the steps that another agent would need to take to get to state A. A non resetting agent would almost necessarily be at a disadvantage in terms of sample complexity if thats how the environment steps were counted.
- Another useful baseline to compare against could be where the agent is reset, but the return is calculated as normal. This could also show whether the max operator is needed.

**Summary Of The Paper:**

This paper introduces a new setting which utilizes resetting to previously visited states in the trajectory and continuing on from there. The objective is then defined as the max return over the resulting trajectory tree. The paper shows that directly optimizing for this objective results in stochastic, exploratory policies.

**Summary Of The Review:**

I think this is an interesting idea, and should eventually be published, but I think it is not ready in its current state. Its contributions need to be reframed, certain baselines need to be added, and the way the experimental results are presented need to be changed.

---

> ### Author Response · Authors · 2022-11-07
> **Response**
>
> Thank you for the review.
>
> >Another useful baseline to compare against could be where the agent is reset, but the return is calculated as normal. This could also show whether the max operator is needed.
>
> Actually, this comparison was performed in appendix C.2 figure 12. The results showed that resetting alone improves the performance; however, using the max operator further improved the performance and was necessary to lead to a stochastic policy. These results were provided in addition to the study in the main paper in section 7 where we compared to the average operator instead of the max. Both of these results confirmed that the max operator was needed. We will modify the manuscript to better highlight the additional ablation study in the appendix.
>
> >The paper compares against other methods that don't reset the simulator during training.
>
> The comparison with the average operator and the separated variants both use resetting, and did not perform as well as using the max operator in ReMax.
>
> >It would be helpful to see how this policy actually works as an exploration policy used to train another policy that is trying to optimize the original RL objective.
>
> Could you elaborate on this point? We clarify that our method improves the exploration for training the current policy, while we evaluate the policy by picking the most likely action deterministically (and evaluate on the original RL objective). This could be seen as using the stochastic policy to train this deterministic policy. This worked effectively. The results also showed that simply optimizing the original RL objective performs poorly, because the behavior policy quickly stops exploring. The objective of our method is to enhance the exploration for the policy that we are training rather than enhance the exploration for an arbitrary different policy.

---

### Author Response · Authors · 2022-11-07
**General response and objective of our paper**

Thank you for the reviews. In the interest of starting the discussion, we have added detailed responses to each review in their attached comments. We aim to update our manuscript by the end of the discussion period to further polish it, but would appreciate an initial review of our responses, and whether these changed the reviewers’ point of view, why or why not, or what would be necessary.

Here, we also briefly summarize the objective of our paper, and how this is achieved.

We  proposed a new method for exploration called ReMax that promotes a stochastic exploratory policy by resetting and maximizing the return across the best path in the trajectory tree. We note that it is non-trivial that this approach will lead to any exploration at all (as it is just greedily maximizing the rewards), and the main objective of the current paper was to establish and explain that the approach does indeed lead to exploration. We did this in several phases of experiments, and in each phase we see that (a) ReMax lead to a stochastic policy (b) the standard RL objective lead to a deterministic policy (c) ReMax greatly outperformed the standard RL objective (note that the methods are evaluated on the standard RL objective with deterministic actions and no resetting, but ReMax promotes exploration during training). Our ablation studies showed that the maximization operator in ReMax was necessary for the emergence of a stochastic policy, and the resetting alone was not sufficient to achieve this. We also saw that ReMax was competitive with the entropy-based approach showing that the achieved performance improvement was non-trivial. What remains unclear is whether ReMax should be preferred over the standard entropy-based approach (as the performance on MinAtar was similar, but note that in the maze task ReMax outperformed entropy by a large margin in Figure 7). However, we note that our approach is novel, and there could be several ways to improve it, e.g., by designing new smart resetting methods (e.g., learned reset policies). We believe our approach is interesting because stochasticity is promoted without an explicit exploration bonus, and we are not aware of any previous works with such a property. In conclusion, we believe we have established a new interesting and viable direction for exploration research.

---

### Author Response · Authors · 2022-11-19
**Update**

We have updated the paper with new experimental results.
In the original submission, the value function was learned by the TD error, but we changed this to use n-step returns so that it is consistent with the original A3C.
We find that in the new implementation the performance on SeaQuest improved a lot.
Moreover, ReMax showed relatively improved performance compared to the other methods (actually, on some tasks the performance dropped for the standard A2C).
In the new version we have also added an ablation study on the maximization in ReMax in the MinAtar experiments (previously we only included such an ablation in the Biased Maze task), which showed that the maximization was necessary for the emergence of exploration, and also provided significantly improved performance on 4/5 tasks.
We hope these new results may improve the evaluation of our submission.

---

### Decision · Program_Chairs · 2023-01-20

**Decision:**

Reject

**Justification For Why Not Higher Score:**

Although the idea is interesting, it misses an in-depth comparison with other similar ideas.
The authors need to improve their paper by providing a better understanding of why the proposed approach is sensible and how it relates to existing ones.

**Justification For Why Not Lower Score:**

N/A

**Metareview: Summary, Strengths And Weaknesses:**

The paper proposes an approach that allows the agent to reset its trajectory to a specific state, by optimizing an alternative objective called ReMax. This approach allows for achieving exploratory behaviors without explicitly adding exploration bonuses to the reward function.
The proposed objective is optimized through policy gradient and tested compared to using the standard RL objective in the MinAtar benchmark.
Although the authors study a relevant problem and propose an interesting solution, the reviewers have raised several concerns that have not been completely solved by the authors' feedback.
In particular, the reviewers have pointed out the lack of proper comparison with other methods in the large literature on exploration in RL.
The reviewers complain about a lack of motivation for the proposed approach and would like to have more insights about the way in which the new objective is able to induce exploratory behavior.
We encourage the authors to consider the reviewers' suggestions while preparing a new version of their paper.

**Summary Of Ac-Reviewer Meeting:**

N/A